Late Cretaceous coprolite from the Opole area (southern Poland) as evidence for a variable diet in shell-crushing shark Ptychodus (Elasmobranchii: Ptychodontidae)

http://orcid.org/0000-0001-9020-4100 Mazurek Dawid 1 2
Antczak Mateusz 1 2 mateusz.antczak@uni.opole.pl
1 Institute of Biology, University of Opole , Opole , Polska
2 European Centre of Palaeontology, University of Opole , Opole , Poland
De Baets Kenneth
Electronic publication date: 2023 Dec 15
Publication date: 2023
Volume: 11
Electronic Location ID: e16598
Received 2023 Apr 13; Accepted 2023 Nov 15
Copyright: © 2023 Mazurek and Antczak
Copyright year: 2023
Copyright holder: Mazurek and Antczak
License: This is an open access article distributed under the terms of the Creative Commons Attribution License, which permits unrestricted use, distribution, reproduction and adaptation in any medium and for any purpose provided that it is properly attributed. For attribution, the original author(s), title, publication source (PeerJ) and either DOI or URL of the article must be cited.
License URL: https://creativecommons.org/licenses/by/4.0/

Keywords: Fossil, Coprolite, Chondrichthyes, Shark, Palaeoecology

Funding: The authors received no funding for this work.

==============================
Background

Coprolites, i.e., fossilized faeces, are an important source of knowledge on the diet and food processing mechanisms in the fossil record. Direct and indirect evidences for the dietary preferences of extinct sharks are rare in the fossil record. The first coprolite attributable to Ptychodus containing prey remains from the European Cretaceous is documented here.

Methods

A coprolite from the Late Cretaceous of Opole (southern Poland) was scanned using micro-computed tomography to show the arrangement of the inclusions. In addition, the cross-section was examined under the SEM/EDS to analyse the microstructure and chemical composition of the inclusions.

Results

Brachiopod shell fragments and foraminiferan shells are recognized and identified among the variously shaped inclusions detected through the performed analysis.

Conclusions

The extinct shell-crushing shark Ptychodus has been identified as the likely producer of the examined coprolite. The presence of brachiopod shell fragments indicates that at least some species of this durophagous predatory shark may have preyed on small benthic elements on the sea bottom.

Introduction

Coprolites, i.e., fossilized faeces, together with consumulites (intestine contents), gastroliths (stomach, or gizzard, stones), and regurgitalites (orally expelled masses) make up the group of ichnofossils known as bromalites (Hunt & Lucas, 2021). These are informative for establishing the diet and food processing style. The major caveat is the uncertainty concerning the specific producer of this kind of fossils. Sometimes, the co-occurrence in the same strata of fossils and faeces, and specific features of the animal linking the coprolite and skeletal material (e.g., size, purported diet), can be used as means to pinpoint, with a certain level of certainty, the most likely producer. This was done for the Late Triassic site of Krasiejów in the Opole area, where small coprolites, containing insect remains, were identified as a product of a co-occurring dinosauromorph Silesaurus opolensis, with the main reasoning based on body sizes and possible diets of the skeletally identified fauna at this locality (Qvarnström et al., 2017, 2019, 2021). The discussion there, however, did not take into account a range of taxa from the site identified thus far only on the basis of dental remains. Shark teeth and coprolites are a common find in Late Cretaceous deposits, including the Turonian-Coniacian of Opole area (Mazurek, 2008). Skeletal fossils consist mainly of isolated teeth, with few finds of an associated dentition or even a single vertebra (pers. obs.). Niedźwiedzki (2005) and Niedźwiedzki & Kalina (2003) are the only authors that have studied the shark fauna of the Opole area in recent years. Niedźwiedzki & Kalina (2003) described from Opole the following taxa: Ptychodus latissimus, P. mammillaris, P. polygyrus, Squalicorax sp., Scapanorhynchus raphiodon, and Paranomotodon angustidens. Niedźwiedzki (2005) listed jointly taxa from localities at Opole and Sudetes area. Apart from those mentioned above, other taxa said to be common were Cretoxyrhina mantelli, Cretolamna appendiculata, Squalicorax falcatus, and Odontaspis subulate, while rare finds included Hexanchus microdon, Synechodus major and Hybodus dentalus. In a popular book (Yazykova, 2022), Niedźwiedzki confirms the presence specifically in the Opole area of Squalicorax falcatus, Cretolamna appendiculata, Cretoxyrhina mantelli, and Odontaspis subulata. These works are supplemented by the collecting efforts of the current authors, whose rich collection preserves Squalicorax falcatus and other lamniforms, Ptychodus spp., as well as a single find of a hexanchiform.

As for coprolites, spiral shark faeces are especially common in clayey marls. Their general presence was already noted by Mazurek (2008) and Hunt et al. (2015). Here, we present and document in detail for the first time one of the coprolites from the Upper Cretaceous of Opole (southern Poland). The specimen was analysed by SEM-EDS and microCT to investigate structure and chemical composition of the inclusions. Based on the shape and prey content of the coprolite and the dietary preferences of the co-occurring ichthyofauna, the coprolite producer was identified and its behaviour was discussed in a palaeoecological context.

Geological setting

Odra II quarry is a working quarry within the city of Opole (southern Poland). The exposed rock sequence starts with clayey marls (Middle Turonian Inoceramus apicalis Zone) and proceeds with limy marlstones (Middle Turonian I. lamarcki Zone to the lowermost part of Upper Turonian I. perplexus Zone), and ends with marly limestones (I. perplexus Zone). This sequence of strata forms part of a single transgression-regression megacycle (Cenomanian-Coniacian) that represents the Cretaceous strata of the so-called Opole Trough (Jagt-Yazykova et al., 2022). The biota preserved is numerous and consists of ichnofossils, sponges, inoceramids and other bivalves, brachiopods, fish remains, cephalopods, echinoderms, crustaceans, cnidarians, shark coprolites, land flora, and rare marine reptiles. The coprolites are quite common and of uniform size and shape, with spiral structure pointing to sharks (Dentzien-Dias et al., 2012). The specimen studied comes from the clayey marls (Middle Turonian: I. apicalis Zone).

Materials and Methods

A coprolite was collected from the Odra II quarry (Oleska street, Opole) during the summer digging camp in 2020. It is housed at University of Opole (col. no. IBUO-DM-KOPRO1). Fieldwork was possible due to the legal agreement between the quarry owner (Cement Factory “Odra”) and European Centre of Palaeontology, University of Opole dated 24.05.2017.

The coprolite is incomplete and the preserved portion is 22 mm in length. The estimated size of the coprolite could be at least two times larger compared to other specimens in the collection ranging between ca. 20–55 mm in total length. As the specimen is broken, some dark infillings are visible within the grey phosphatic mass on the cross-section (Fig. 1). To determine the composition of the infilling, the specimen was analysed with micro CT scanner SkyScan 1273 in Bruker Laboratory in Kontich, Belgium. Obtained data were presented using DataViewer (for multiple cross sections in three directions) and CTVox (for the presentation of the 3D orientation of infillings) software. A 8.5 µm resolution scan is available in the Morphosource database (https://doi.org/10.17602/M2/M514300) in the form of 2,882 TIFF image series.

Figure 1 Ptychodus remains from opole cretaceous.

Analysed coprolite IBUO-DM-KOPRO1 in lateral view (A) and cross-section (B). Coprolite IBUO-DM-KOPRO2 in lateral view (C). Teeth IBUO-DM-ZAB1 (D).

For chemical identification of the infilling, the surface of the broken part (cross-section) was polished with grinding powder. The obtained polished surface was examined under Scanning Electron Microscope TM 3000 with secondary electrons as well as with the use of Energy-Dispersive X-ray Spectroscopy. In addition, the coprolite IBUODM-KOPRO2 (Fig. 1C) was selected as comparison material.

Results

Examined specimen and additional IBUO-DM-KOPRO2 possess a heteropolar spiral shape (see also Fig. 2D), which is typical of chondrichthyan coprolites (see Eriksson et al., 2011; Dentzien-Dias et al., 2012). MicroCT scans reveals numerous infillings with densities differing from the phosphatic matrix of the coprolite (Figs. 2 and 3). Most of the shapes are irregular, many being boat-shaped. Some of them can be recognized and assigned to certain groups of animals, specifically micromorphic brachiopods (Fig. 4) and foraminifera (Fig. 2F), based on, SEM observations of microstructure and cross-section visible in micro CT scan. Two unidentified shells/tests have been observed under higher magnification under SEM. Both inclusions (Fig. 4) show the walls consisting of horizontal lamellae. No vertical elements are present, which would be expected in the case of an inoceramid prismatic layer (e.g., Jiménez-Berrocoso, Olivero & Elorza, 2006), one of the possible prey. No macroscopic chunks of large bivalves are present either. The microstructure is more reminiscent of an inpuncate brachiopod shells (Griesshaber et al., 2007). Regardless, some inclusions are firmly identified as brachiopods and forams (Figs. 2 and 3), while no traces of other possible shelled (e.g., inoceramids, see Hattin, 1975) or soft-bodied prey were detected.

Figure 2 MicroCT scan of the coprolite.

Infillings–3D model (A and B). Coprolite mass with infillings–3D model (C and D). Longitudinal cross-section (E and F). b, brachiopod shell; f, foram shell. S, spiral structure. Gavelienella illustration from Hornibrook, Brazier & Strong (1989), Fig. 18.17. Brachiopod shell photograph from alexstrekeisen.it. 3D model made in CTVox. Scan resolution: 8.5 µm. Image credit: https://pal.gns.cri.nz/foraminifera/www/HBS362.htm, © Copyright in 2018 by GNS Science and is licenced for re-use under a Creative Commons Attribution 4.0 International License.

Figure 3 Cross sections of the analysed coprolite.

Cross-sections of the analysed coprolite in three directions (A, C and D). Magnification of the example of indet. shell fragment (B). Image obtained in DataViewer.

Figure 4 EDS analysis.

Brachiopod shell fragments (A and B), the surface of the EDS analysis (C), and mass percentage result (D). SEM photographs: the authors. They were made at Faculty of Chemistry, University of Opole, Opole, Poland.

In the EDS analysis, the main elements are Ca, O, C, and P (Fig. 4).

Discussion

Irregular and boat-shaped infilling creates a specific pattern. Similar infillings can be observed in coprolites of durophagous fishes from the Middle Triassic (Antczak et al., 2020). EDS signature suggests that these are elements made of calcium carbonate. The matrix of the coprolite possesses a phosphatic character. The spiral heteropolar nature of the Opole Cretaceous coprolites points to sharks as their producers. Taking into account the above, it strongly suggests that the analysed coprolite was produced not by a piscivorous shark but rather by species feeding on invertebrates with calcareous shells. The only known candidate is Ptychodus. Currently, this genus is thought to be a facultative durophage, with diet composed of inoceramids and other shelly fauna, but also fishes (Shimada, Rigsby & Kim, 2009; Amadori et al., 2019, 2020, 2023; Hamm, 2020). The assignment of some of the infillings to brachiopods suggests that the producer was feeding at the bottom of the sea (nektobenthonic) instead of in open water (nektonic). In addition, tests of calcareous foraminifera can be recognized, similar to genera Lenticulina or Gavelinella (Kłapciński & Teisseyre, 1981; Strong, Raine & Terezow, 2018) which are bottom-dwelling taxa, probably swallowed accidentally together with the sediment and a brachiopod laying on the bottom of the sea.

In the Turonian of Opole, several shark species could produce coprolites of this size. These include: Cretoxyrinha, Scapanorhynchus, Hexanchus, Squalicorax, and Ptychodus. Among them, only the last is commonly described as durophagous based on tooth morphology (Shimada, Rigsby & Kim, 2009; Shimada et al., 2010; Amadori et al., 2022) (Fig. 5). Niedźwiedzki & Kalina (2003) identified at the Opole Cretaceous three taxa of Ptychodus. Apart from isolated teeth, the Opole Cretaceous also yielded two sets of teeth: one is deposited at the University of Wrocław (MGUWr, unnumbered), while the other is in collection of the University of Opole (IBUO-DM, unnumbered). Similar finds are known for several taxa worldwide (Amadori et al., 2019; Hamm, 2017), with partial skeletons or skulls much rarer (Shimada, Rigsby & Kim, 2009; Shimada et al., 2010).

Figure 5 Ptychodus reconstruction (Author: Jakub Kowalski) with an example of tooth IBUO-DM-ZAB1 and coprolite (IBUO-DM-KOPRO2).

Magnification of the internal structure of the coprolite comes from IBUO-DM-KOPRO1.

The occurrence of Ptychodus as the only durophagous shark suggests that the producer of the coprolite might be specifically identified to the mentioned genus. However, the lack of bivalve shell fragments within the coprolite is notable. There are several possible explanations.

The first possibility is that the producer of the coprolite fed also on the common inoceramids, but was able to feed only on the soft tissue and for example orally reject the hard shells. The modern mammal Odobenus rosmaris feeds on benthic mollusks by sucking the soft tissue and ejecting the hard parts (Scheyer et al., 2011). However, currently, no dentalites were recognized from Opole Cretaceous inoceramid shells (even though many microscopical epifauna remnants can be observed—e.g., Bryozoa, Serpulidae, Ostreoida). From numerous specimens described by Walaszczyk (1992) a single sublethal injury was mentioned. If sharks were efficient predators we would predict evidence of failed prey subjugation. However deformations and growth iterations in inoceramid shells are known, they are, rather, effects of decapod predation (Harries & Ozanne, 1998). Of note, none of the coprolites we studied externally seem to contain any large shelly material. To the best of our knowledge, none are known elsewhere.

The second possibility is that the fossils of a coprolite producer are not present (or not recognized yet) in the Quarry due to the sedimentation bias or being less common representative of the Cretaceous fauna of this area. Hunt et al. (2015) show that producers of coprolites are often not represented by body fossils. Chondrichthyan fossilized faeces are the most common, while in terms of body fossils palaeoichthyofaunas are usually much more diversified, which Hunt et al. (2015) termed the ‘shark surplus paradox’.

The third option, explaining this the lack of dentalites and brachiopod infillings in the described coprolite, is to consider Ptychodus (the form from Opole, and by extensions possibly also other members of the genus) as the producer which, contrary to some current opinions, was not a strictly durophagous taxon, but rather a durophagous-filter feeder specialized in small prey, with bulbous teeth for crushing shells, but also with water moving between the ridges of the teeth (Fig. 1) or more likely rejecting water and sediment through gills like modern myliobatiform rays that fluidize the sediment by means of jaws’ movement (Sasko et al., 2006). The sediment of the Cretaceous chalk seas might already be soupy in consistence and Ptychodus might sift it in search for small shelly fauna. Such elaborated ornamentation as present on the teeth of Ptychodus is lacking in many other durophagous taxa except skates, including among others: various fishes (e.g., Purnell & Darras, 2015; Raguin et al., 2020), placodonts (Pommery et al., 2021) and many mosasaurs (Leblanc, Mohr & Caldwell, 2019); the teeth are often restricted to the outer edge of the jaws, and supposed shark dentalites on inoceramids and other hard elements are rare in the literature known to us (e.g., Kauffman, 1972; Hunt & Lucas, 2021, Table A.5), which, however, can be ascribed to poor taphonomic potential of such finds, and the lack of both recognition and studies devoted to them. Also, not all filter-feeders possess small, gracile, sieve-like teeth. Several species of pinnipeds have teeth modified into filter-feeding, specifically with elaborate cusps of postcanines on both the upper and lower jaw. This modification is well-seen, especially in the crabeater seal Carinophaga lobodon (Chatterjee & Small, 1989; Bengtson, 2002; Adam, 2005).

Conclusions

MicroCT scan and EDS analysis show that coprolite collected in the Turonian deposits of Odra II quarry in Opole, southern Poland is filled with shell fragments. Inclusions can be identified as remains of small brachiopods (and occasionally Foraminifera). Such content suggest that the producer’s diet was based on the small shell-covered organisms encased within the sediment, possibly revealing mix of durophagy and filter-feeding strategy, i.e., a process of sifting the sediment first and then crushing the remaining fraction. According to the shape of the coprolite, it can be described as belonging to shark. Within chondrichthyan fauna of the locality, there is only one species of durophagous shark, Ptychodus; thus, it can be proposed as the likely producer of the analysed coprolite, although the impact of the ‘shark surplus paradox’ (the high diversity of ichthyofaunas contrasting with a low diversity of coprolite ichnofaunas in Cretaceous chalk facies) cannot be entirely ruled out (Hunt et al., 2015).

Ptychodus (if considered a producer) might have been a durophagous-filter feeder (partially analogous to modern myliobatiforms feeding habit) and not a strictly durophagous fish as there is no evidence of preying on abundant large inoceramids and other common shelly organisms (in the forms of coprolites or regurgitalites). While we acknowledge this hypothesis cannot necessarily be universally applied to other species of the genus, or different growth stages—in the context of scarcity of direct evidence worldwide for preying on large shelly organisms, we tentatively suggest that some form of both durophagy and filter feeding ecology might need to be considered for Ptychodus spp. individuals. Further investigation of coprolites and, when available, gut contents will be necessary to confirm or reject the hypotheses proposed in this study.

We would like to thank Piotr Czerwiński and the COMEF company for the possibility to scan the specimen in the Bruker Laboratory in Kontich and for providing the software to present the data. We are grateful to Wioletta Ochędzan-Siodłak for the possibility to use SEM/EDS at the Faculty of Chemistry (University of Opole) and technical help with the analysis. We also want to thank Elena Yazykova for the many fruitful discussions and the overall supervision of works done in the Opole Cretaceous and Jakub Kowalski for the drawing of Ptychodus. Sincere thanks are also due to the Reviewers (Adrian Hunt, Manuel Amadori, Hannah Byrne) and Editor (Kenneth De Baets) for many important comments and advice that greatly helped to improve the manuscript and to make the presented hypothesis consistent.

Additional Information and Declarations

Competing Interests

Author Contributions

Field Study Permissions

Data Availability

The authors declare that they have no competing interests.

Dawid Mazurek conceived and designed the experiments, performed the experiments, analyzed the data, prepared figures and/or tables, authored or reviewed drafts of the article, and approved the final draft.

Mateusz Antczak conceived and designed the experiments, performed the experiments, analyzed the data, prepared figures and/or tables, authored or reviewed drafts of the article, and approved the final draft.

The following information was supplied relating to field study approvals (i.e., approving body and any reference numbers):

Field works were approved by the quarry owner on the rights of agreement between Odra S. A, and European Centre of Palaeontology (24.05.2017).

The following information was supplied regarding data availability:

The CT scan is available at Morphosource: https://doi.org/10.17602/M2/M514300.

The physical specimen is housed at University of Opole palaeontological collection (IBUO-DMKOPRO1).

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
