# Peer review of "Late Cretaceous coprolite from the Opole area (southern Poland) as evidence for a variable diet in shell-crushing shark Ptychodus (Elasmobranchii: Ptychodontidae)"

_PeerJ, doi:10.7717/peerj.16598_

## Round 0.1 · original submission · Major Revisions

Thank you for this interesting coprolite discovery and its interpretation allowing to attribute it to its producer and potential implications for its diet. As only rarely coprolites can be attributed precisely, it would love to see it published. However, some crucial aspects need to be addressed before publication:

Title: I feel the current title is unnecessarily provocative and not entirely correct (e.g., the assignment of the producer remains uncertain; also compare reviewers 2 and 3). Also, I feel such a title not necessary as your study is relevant and would benefit from absence of such a contentious title. My suggestion would be to rename it to something along the lines of: Whose poop is this? Late Cretaceous coprolite suggests a more variable diet of Ptychodus. (further modifying the suggestion made by reviewer 3 taking into consideration comments by the other reviewers on your interpretations).

Computed tomography: Thank you for making available and uploading raw data on morphosource already during the review process. It is however essential to provide the final 3D models used in the study (e.g., STL) formats than can be opened by everyone for the sake of scientific reproducibility at the latest upon publication on such a platform as well as provide additional information or meta-data on the conditions scanning in the manuscript (e.g., used Energy, filters) in addition to the full resolution image stack (see Davies et al. 2017). This 3D-models were not relevant yet as you only seem to have worked with raw scans, but I agree with reviewer 3 that your analysis would be more convincing with 3D reconstruction of the coprolite particularly its inclusions.

Resolution of provided photographic documentation and illustrations: the currently provided picture and other material is quite low in resolution and focus. Also scale bars are necessary (compare reviewer 2). In addition, computed tomographic documentation need further documentation for your arguments to be convincing (compare reviewer 3).

Attribution to Ptychodus: Although I agree that Ptychodus seems like its most likely producer if no other benthic, durophagous producers are documented from Opole, this is not certain (maybe it belongs to a rare not yet described predator or one not even represented in the fossil record; compare reviewer 1). You need to document the arguments more clearly for assigning it to a benthic durophagous predator (e.g., better documentation of prey items, potentially inclusion of sediment) as well as an elasmobranch producer (e.g., spiral morphology of the coprolite and ideally teeth inclusions although I knowledge the latter are likely rare: compare reviewer 2 and 3).

Documentation of (benthic) inclusions: Another crucial argument to assign it to a demersal or benthos-feeding producer are the inclusions. Your arguments would be more convincing if you would provide actual 3D reconstructions of the prey items rather than virtual sections (compare reviewer 3).

Documentation of spiral structure: A crucial argument for an elasmobranch producer is its type of spiral structure (e.g., Dentzien-Dias et al. 2012, 2021; Rodrigues et al. 2018). As it is a crucial argument to assign it to larger group, you need provide a better documentation of the morphology and type of spiral structure of the coprolite (if a section is not possible and parts are missing), you need to document/show this structure in a well-resolved CT scan. It is not straight forward as some diagenetic alterations can give the impression of a pseudospiral structure which can superficially be confused with true spiral structure. Other (more) complete coprolites with similar morphology (e.g., morphotype) from the same locality might also provide crucial additional information and context.

Change in diet of Ptychodus: I agree with reviewer 2, that your argumentation can currently be seen as circular and should be more clearly designated as speculative. Part will remain speculative without more direct evidence for the assignment of the coprolite to Ptychodus (e.g., a broken teeth within the coprolite: compare Shimada 1997, characteristic of coprolite spiral morphology specific to this lineage which might be hard to find) or other supporting evidence for a direct interaction between Ptychodus and (brachiopod) prey items. The latter might be hard to find when it is rather a “filter” or “deposit” feeder as opposed to a durophagous shell crusher but an absence of observations in the fossil record is not necessarily a proof of absence (e.g., Zapalski 2011). My suggestion would be to structure the manuscript more clearly in assigning the coprolite to its producer (based on morphology of the coprolite and comparative analysis with other coprolites assigned to hybodont sharks; the degree of its inclusions being benthic and previously recovered from such types of coprolites and comparisons with other types of larger benthic predators capable of handling durophagous prey items). The possible uncertainty in assigning the coprolite should be made clearer but I do not see this as a hindrance to the publication or relevance of your paper when all aspects are better documented and argumented/discussed in greater detail. In this context, it might also make sense to discuss other cases of assignment of a diverse diet to hybodontid sharks based on damage on prey items (e.g., Kauffman 1972; Klug et al. 2021), stomach contents (e.g., Maisey and Carvalho 1997; Doyle and Macdonald 1993; Klug et al. 2021) and the association or correlation characteristics of particular dentition with prey items (Shimada 1997; Vullo and Néraudea 2008; Vullo 2011). I also agree with reviewer 2 that you also need to provide additional support for the established diet and its potential variability in Ptychodus based on published literature (e.g., published wear on teeth, correlation with teeth morphology and contents in fossil assignable to the digestive system if now at all or other (related or unrelated) taxa where it is the case). In this context, a more detailed depiction of the teeth of Ptychodus would also be helpful (compare reviewers 1,2 and 3).

Reconstructions of producer and coprolites: I feel adding a reconstruction of Ptychodus could be a nice addition, but it must be reworked based on our known knowledge about their anatomy (compare reviewer 2) and the degree of speculation should be more clearly indicated – I would consider a benthic “deposit” feeder to look different from a pelagic “filter” feeder. More importantly, I feel it would be crucial to add a reconstruction of the coprolite complete with its possible reconstruction (supported by arguments as to why you think the complete coprolite should look like this) and this could be done in the same figure or by adding an additional or even replacing this one (if you prefer).

Definition of filter-feeding: you need to provide a definition of what you consider filter feeding – ideally based on previously published definitions. To my knowledge, most definitions and research has focused on lineages which are more pelagic in assigning this type of diet. It is clear there are some vertebrate lineages might potentially include smaller benthic items in their diet, but the question remains if these should be considered deposit rather than filter-feeding. Given that you only document one (accidental?) foraminifer in addition to multiple crushed fragments of (multiple?) brachiopods, a benthic, durophagous diet might make more sense in your case. In addition, specialized predators might be less or more adept in avoiding hard “shell” fragments to end of in their digestive systems (see Reviewer 1; see also Klug et al. 2021 for the discussion of relevance of belemnite rostra or fragments in stomach contents or consumulites more generally versus lack thereof)


Typographical and formatting issues: there are some issues which need attention (e.g., faeces should be written consistently, “Concluions” should be “Conclusions” in abstract). I did not find any additional issues then the ones already pointed out by the reviewers.

Cited references: you nicely reference the available relevant literature in most cases (e.g., the Opole site) but additional supporting literature is necessary for the definition of bromalites (compare reviewers 1 and 3) as well as established diet of Ptychodus and other elasmobranch taxa (compare reviewer 2 and 3).

Please address these as well as all other points raised including those in reviews and annotated pdfs.

I look forward to receiving your revised manuscript.

Suggested references:

Davies, T. G., Rahman, I. A., Lautenschlager, S., Cunningham, J. A., Asher, R. J., Barrett, P. M., ... & Donoghue, P. C. (2017). Open data and digital morphology. Proceedings of the Royal Society B: Biological Sciences, 284(1852), 20170194.

Dentzien-Dias, P. C., de Figueiredo, A. E. Q., Horn, B., Cisneros, J. C., & Schultz, C. L. (2012). Paleobiology of a unique vertebrate coprolites concentration from Rio do Rasto formation (Middle/Upper Permian), Paraná Basin, Brazil. Journal of South American Earth Sciences, 40, 53-62.

Dentzien-Dias, P., Hunt, A. P., Lucas, S. G., Francischini, H., & Gulotta, M. (2021). Coprolites from shallow marine deposits of the Nanjemoy Formation, Lower Eocene of Virginia, USA. Lethaia, 54(1), 26-39.

Doyle, P., & Macdonald, D. I. M. (1993). Belemnite battlefields. Lethaia, 26, 65–80.

Klug, C., Schweigert, G., Hoffmann, R., Weis, R., & De Baets, K. (2021). Fossilized leftover falls as sources of palaeoecological data: a ‘pabulite’comprising a crustacean, a belemnite and a vertebrate from the Early Jurassic Posidonia Shale. Swiss Journal of Palaeontology, 140(1), 10.

Maisey, J. G., & de Carvalho, M. R. (1997). A new look at old sharks. Nature, 385(6619), 779-780.

Rodrigues, M. I. C., da Silva, J. H., Santos, F. E. P., Dentzien-Dias, P., Cisneros, J. C., de Menezes, A. S., ... & Viana, B. C. (2018). Physicochemical analysis of Permian coprolites from Brazil. Spectrochimica Acta Part A: Molecular and Biomolecular Spectroscopy, 189, 93-99.

Shimada, K. (1997). Shark-tooth-bearing coprolite from the Carlile Shale (Upper Cretaceous), Ellis County, Kansas. Transactions of the Kansas Academy of Science (1903), 133-138.

Vullo, R. (2011). Direct evidence of hybodont shark predation on Late Jurassic ammonites. Naturwissenschaften, 98(6), 545-549.

Vullo, R., & Néraudeuau, D. (2008). When the “primitive” shark Tribodus (Hybodontiformes) meets the “modern” ray Pseudohypolophus (Rajiformes): the unique co-occurrence of these two durophagous Cretaceous selachians in Charentes (SW France). Acta Geologica Polonica, 58(2), 249-255.

Zapalski, M. K. (2011). Is absence of proof a proof of absence? Comments on commensalism. Palaeogeography, Palaeoclimatology, Palaeoecology, 302(3-4), 484-488.

·

Basic reporting

The manuscript in written in clear English. I have made some minor edits to improve the English.
The Introduction is good – I have made a suggestion about using a more precise terminology (consumulite rather than cololite – cololite refers only to contents of intestines and consumulite refers to all contents of any part of the GI tract) and provided a relevant reference (Hunt, A. P. and Lucas, S. G., 2021, The ichnology of vertebrate consumption: Dentalites, gastroliths and bromalites: New Mexico Museum of Natural History and Science Bulletin 87, 215 p.).
The structure of the paper is sound, but I would move the geology section (last paragraph of Materials & Methods) to its own section for clarity.
The figures are appropriate.

Experimental design

The primary research is within the scope of the journal. The basic research question is well defined as to identifying the producer of the coprolite and how this may provide information on the feeding of Ptychodus. The methodology is appropriate and well described.

Validity of the findings

The idea that Ptychodus was not purely durophagous is certainly worth considering but to justify such a significant new interpretation I think there should be more discussion, for example in regards to:

1. Why would a filter feeder have large bulbous teeth? Are there any analogues? It would be interesting to compare with the work of Diedrich who tried to argue that placodonts were not durophages but rather herbivores (e.g., . Scheyer, Torsten M., James M. Neenan, Silvio Renesto, Franco Saller, Hans Hagdorn, Heinz Furrer, Olivier Rieppel, and Andrea Tintori. "Revised paleoecology of placodonts–with a comment on ‘The shallow marine placodont Cyamodus of the central European Germanic Basin: its evolution, paleobiogeography and paleoecology’by CG Diedrich (Historical Biology, iFirst article, 2011, 1–19, doi: 10.1080/08912963.2011. 575938)." Historical Biology 24, no. 3 (2012): 257-267.
2. The coprolite could have been produced by an animal not represented by body fossils. The coprolite ichnofauna does not always represent the body fossil fauna (see the discussion in Hunt et al., 2015 that is cited in the manuscript).
3. Just because Ptychodus may have fed on inoceramids does this mean that one would expect large shell fragments in its feces – they might well have been regurgitated rather than digested?

Additional comments

I think this is a very interesting study of an unusual coprolite. I commend the authors for presenting such an unexpected new hypothesis for the feeding habits of the abundant taxon Ptychodus.

·

Basic reporting

Overall, the submitted study is interesting and the idea of using coprolites and other ichnofossils for hypothesizing dietary preferences in extinct taxa is intriguing. The hypothesis of Ptychodus being a filter feeder is also fascinating. While “significantly brief, concise and compact”, the presented manuscript is written in proper and correct English with a few sporadic spelling mistakes (e.g., “Concluions” in the abstract). However, this reviewer is afraid the major issues of the submitted study are in its contents rather than in drafting and writing style. The approaches, methods and analysis protocols used in it seem well designed and adequate to ensure solid and rigorous results. However, the general structure of the study, and consequently of the manuscript documenting it, is very confusing, unclear and does not meet the standards required for a modern scientific publication. In the following text, this reviewer explains in detail lacks, flaws and issues concerning the submitted manuscript.
In its current form, the title “Whose poop is this? The case of the Late Cretaceous coprolite with clearly identified producer and the feeding habit of Ptychodus” is catchy but inappropriate for a scientific publication. One might gloss over the funny opening question "Whose poop is this?”, which, on the other hands, anticipates the real problem within the submitted study. Nevertheless, even ignoring the opening, the chosen title is misleading and definitely out of place by confidently claiming a “clear attribution” of the described coprolite to Ptychodus. In the opinion of this reviewer, this is indeed far from obvious and poorly supported by the results presented within the submitted manuscript (see point 3 of this review, below).
The “very compact” introductive section presents briefly the main topic of the submitted study: coprolites. The relevance of the subject “for establishing the diet and food processing style” (see line 34) is also shortly mentioned. In addition, a few lines of the submitted introduction are also dedicated on major issues and uncertainties especially related to the identification of the possible producer of coprolites (see line 35). Although the current content of the introductive section is relatively well written, it is too brief and lacking a solid background. Indeed, a proper introduction on the state of the art for the study of coprolites (and other ichnofossils), which would have provided the perfect framework for this study, is missing in the submitted text. In particular, a more comprehensive introduction would have ensured a full understanding of not only valuable insights derived from the study of ichnofossils (here coprolites), but also doubts, issues and concerns related to their investigation. Moreover, an appropriate introduction on the group of fossil sharks assumed to have produced the examined coprolite (here Ptychodus) with special focus on feeding mechanisms and previously hypothesized diet preferences would have been necessary, more than just desirable, for the presented study. Moreover, a few more references (especially for the main aspects involved in the study of coprolites) would have been desirable and appreciated. Referring to relevant literature for core and fundamental concepts and terminology should be indeed common practice in writing a scientific paper.
For the above, the introduction as a whole is therefore to be considered incomplete and inadequate for publication, in the opinion of this reviewer.
The four figures attached to the submitted manuscript could have been sufficient in quantity, details displayed and information provided. Unfortunately, the quality of the images is inappropriate for a palaeontological study especially. The lack of scale bars or any measurement reference in figures 1a, 2 and 3 (see submitted manuscript) is unacceptable in any modern scientific publication. Even when scale bars are included within the images, the scale values are not indicated in the submitted captions. The captions are short, but sufficiently explanatory. This reviewer is also confused by the “Ptychodus reconstruction” in figure 5, whose relevance to this study remains unclear anyway. The presence of dorsal spines and cephalic spines in the submitted life reconstruction of Ptychodus are indeed very ambiguous as they are usually characteristic of hybodont sharks. Ptychodus is unlikely to belong to the latter given the calcified vertebral centra occurring in its fossil record, which are lacking instead in the palaeontological record of hybodonts. Moreover, based on the hypothesis of filter feeding for Ptychodus proposed in the submitted study, this author would have expected a life reconstruction more closely resembling a filter-feeding shark (e.g., whale shark) than a weird mixture between a hybodont and a sort of nurse shark.

Experimental design

As mentioned above, no particular concerns are related to the methodologies employed for the submitted study. In particular, the use of multiple combined microscopy techniques (here CT-scan and SEM/EDS) is certainly the best approach to efficiently evaluate fossil material of difficult interpretation, such as ichnofossils (e.g., coprolites). Modern analysis approaches and protocols applied to the analysis performed for the submitted study are therefore to be considered valid and appropriated for a scientific publication.
The 'Materials & Methods' section is concise, but clear, informative and written in proper English. Therefore, there are no particular faults or shortcomings to report in this regard.

Validity of the findings

Surprisingly, the authors do not explicitly and clearly state anywhere in the manuscript the primary aim and research questions underlying the submitted study. However, “it seems quite evident that” the purpose of the submitted study is to infer a new feeding habit (e.g., see title) for extinct sharks belonging to the genus Ptychodus based on fossilised faecal remains (coprolites). Establishing feeding and diet range of extinct organisms is often very arduous given the scarcity of direct evidences of prey-predator interaction within the fossil record (e.g., Kelley et al. 2003). This is an even more challenging task if the study group (here, sharks) is nowadays composed almost exclusively of generalist predators notoriously feeding on a wide range of prey (e.g., Bels and Whishaw 2019). A significant scientific impact is therefore expected for any study, including the submitted one, which can provide new insights on feeding and predation for extinct organisms. The original core idea of the submitted work therefore matches the scopes of the journal chosen for publication (“PeerJ - the Journal of Life & Environmental Sciences”).
The “concise and brief writing style” that generally characterises the proposed manuscript is also found in the “Results” section. The obtained outcomes are nonetheless described quite clearly within the provided text. The rest of the material submitted to support the hypotheses proposed by the authors (figures) are insufficient and inappropriate (see point 1 of this review, above).
Despite good results were obtained from the analyses performed for the submitted study, serious issues must be highlighted within the data interpretation proposed by the authors. In contrast to the previous sections of the submitted manuscript, the “Discussion” section is very confusing and lacks clear and solid reasoning. This reviewer had serious difficulties understanding the argumentation, although written in good English, behind the proposed interpretations. In particular, two major issues must highlighted in the “Discussion”: (A) the “clear” attribution of the coprolite to Ptychodus and (B) the “well-supported” new feeding (filter feeding) for the genus Ptychodus. These two key points are further discussed in detail in the following text:
A) As far as this reviewer could understand from the submitted manuscript, the attribution of the examined coprolite to Ptychodus was based on the similarity between the “infilling pattern” observed in the analysed specimen and those found in coprolites associated to shell-crusher fishes (durophagous) from the Middle Triassic. Ptychodus is indeed the only sharks adapted to crush shelled prey (durophagy) known from the Turonian of the Opole area; this is clearly stated in the submitted manuscript (see lines 117-119). The argument presented by the authors to support the attribution to Ptychodus is surprising and contradictory to say the least. The authors themselves indeed attribute the examined coprolite to a filter-feeding shark (see lines 125-128), rather than a durophagous predator, and thus propose a new feeding habit for Ptychodus. Based on this rather contradictory reasoning, legitimate questions spring to mind: do the authors consider the coprolite producer (here Ptychodus) a filter feeder, a durophagous or both? If the coprolite is attributed to a filter-feeding shark, how can we be sure that this is actually Ptychodus, which is so far well known to be a durophagous predator (see also point B, below), and not another filter-feeding fish? If the coprolite producer is a filter feeder, can we be sure that the “infilling pattern”, also associated with durophagous fishes, is indicative of a specific group? If the coprolite is attributed to a durophagous, on what basis can we propose a new feeding habit for Ptychodus?
In the absence of concrete answers to these questions, this reviewer considers the attribution of the examined coprolite to Ptychodus as unreliable.
B) Even blindly assuming the producer of the analysed coprolite to be Ptychodus, a number of further doubts and concerns regarding the proposed new feeding hypothesis follow the already questionable attribution. The content of the examined coprolite mostly includes brachiopod shells and foraminifers, while inoceramid bivalves are missing (see lines 23-25, 101-103 and fig. 2, 3). Based on the content of this single, incomplete coprolite, the authors propose a new feeding habit (filter feeding) for Ptychodus (see lines 126-128, 140-143). Once again, the reasoning presented by the authors are quite confusing and dubious. According to various authors (e.g., Shimada 2012; Everhart 2017; Amadori et al. 2019, 2020, 2022; Hamm 2020), Ptychodus was very diverse group of sharks able to hunt, grasp and process a large variety of prey (from small fishes to hard-shelled mollusc). However, the molariform teeth characterizing the dentitions in Ptychodus suggest its primary adaptation to durophagy (e.g., Amadori et al. 2020; Hamm 2020). Furthermore, occlusal abrasions were commonly described and associated with shelled prey processing for these extinct predatory sharks (e.g., Niedźwiedzki & Kalina 2003; Shimada 2012; Amadori et al. 2019, 2020, 2022; Hamm 2020). Therefore, we are facing an already complex ecological scenario of difficult interpretation for this extinct group of sharks. In general, the idea of shedding light on the feeding of an extinct generalist shark based on partial fecal remains (coprolites) seems interesting to this reviewer but a little too ambitious for the specific case study (Ptychodus). The analysed coprolite contains no bivalves, but still shell fragments from benthic brachiopods. We cannot thus totally exclude the possibility that the producer was a durophagous predator (e.g., Ptychodus). Based on the size reported by the authors (22 mm) for the incomplete coprolite (see lines 71-72), a juvenile individual could have produced it after crushing and feeding on small brachiopods on the sea bottom. This would also explain more convincingly the preserved state of the brachiopods (fragmentary). Furthermore, the distribution of the food remains preserved within coprolites produced by vertebrate (e.g., sharks) in rarely homogeneous (e.g., Eriksson et al. 2011; Hunt et al. 2012; Schwimmer et al. 2015). The specimen analysed is broken (see lines 71-72) and the complete content of the analysed coprolite, possibly including other shelled prey, remains thus unknown. Even assuming Ptychodus was a filter feeder, one would still have to explain the marked dental abrasion commonly documented on teeth of all the species reported from the Turonian of Opole (P. latissimus, P. mammillaris and P. polygyrus; see Niedźwiedzki & Kalina 2003; Amadori et al. 2020; Hamm 2020). This reviewer is keen to point out that he does not rule out the fascinating possibility that at least some species (e.g., cuspidate) within the genus Ptychodus could have used filter feeding as an additional trophic strategy. For examples, a certain degree of apical and lingual abrasion on cuspidate teeth of marine mammal predators, which use filter feeding as an additional predation strategy, was sporadically described (e.g., Hocking et al. 2013; Marx et al 2016). However, the absence of solid evidence within the submitted manuscript to support this filter-feeding strategy for Ptychodus must be highlighted and taken under consideration here. In addition, the alleged absence of Ptychodus-like ornamentations on the tooth crowns of other durophagous groups claimed by the authors (see lines 128-133) is surprising at the very least. For instance, some shell-crusher batoids, such as Rhina ancylostoma (bowmouth guitarfish; see Berkovitz & Shellis 2017), have tooth ornamentations very similar to those described for Ptychodus polygyrus and P. latissimus (Niedźwiedzki & Kalina 2003; Amadori et al. 2020; Hamm 2020). Furthermore, the teeth of many extent and extinct durophagous batoids are usually arranged in wide tooth plates as in Ptychodus (Berkovitz & Shellis 2017). This contradicts another curious theory advanced by the authors stating that teeth of durophagous predator are usually restricted to the edges of the jaws (see lines 131-133). The authors also reported that the coprolite producer was probably feeding at the bottom of the sea instead of in open marine environment (see lines 111-113). Conversely, nowadays filter-feeding sharks mainly feed along the water column rather than on the sea bottom (see Ebert et al. 2021). This provides a further inconsistency between the established lifestyle (nektobenthonic) of the coprolite producer and the feeding hypothesis proposed for it. The authors correctly point out that direct evidence of durophagous shark bites on inoceramid bivalves is rare (see lines 132-133) in order to support their new feeding hypothesis. Unfortunately, this can also be stated for numerous other trophic strategies, filter feeding included, as direct evidences of prey-predator interactions as notoriously scarce and / or dubious within the fossil record (e.g., Kelley et al. 2003). In addition, the more obvious food remains resulting from shell-breaking predation by durophagous specialists would be shell fragments, which can be easily confused with those induced by abiotic breakage (Kelley et al. 2003). Therefore, direct evidences of durophagous predation are in general difficult to recognised and identified.
For all the above, discussions and conclusions provided within the submitted manuscript are to be considered flawed and not supported by solid scientific evidences.

Additional comments

The criticism in this review is not intended to demean the work done by the authors, but rather to point out the major weaknesses and flaws in the resulting manuscript.
The negative assessment provided by this reviewer on most of the material provided (text and figures) leads to the suggestion of rejection for the submitted manuscript; this indeed would require a total reworking to make it suitable for publication. In particular, a better-structured manuscript with broader introduction, results documented in greater detail, well-supported interpretations, solid conclusions and figures of higher quality would be just a good starting point.
Although the presence of Ptychodus from the Cretaceous of the Opole area has already been confirmed, little is still known about the Ptychodus faunas from this region. Therefore, the publication of fossil material certainly assignable to Ptychodus, such as isolated and/or associated teeth, from the Opole area is strongly encouraged by this reviewer. In this perspective, this reviewer will be available and glad to review future manuscripts from the authors and, possibly, provide positive feedback for their further submissions. This reviewer wishes the authors all the best for their forthcoming studies.
A complete list of the references used for this review is in the attachments.

Sincerely,

Dr. Manuel Amadori

·

Basic reporting

Clear and unambiguous, professional English used throughout.
The English is generally in good order throughout the manuscript, however, it could be improved in the abstract. I suggest you have a colleague who is proficient in English to reword the abstract section.
Example - Line 17: Spelling of faeces should remain consistent throughout manuscript, “an” should be inserted between are and important.

Literature references, sufficient field background/context provided.
The manuscript is filled with many relevant references. The Introduction could do with some extra references i.e. Lines 32-34 where the authors state that the different kind of bromalites, they should reference a large review paper like Qvarnström et al 2016 or Hunt and Lucas 2012.
The author’s have provided a good overview in the introduction and great detail on the fossil record from the Opole area. This section would be further improved with an addition to the end stating they have a coprolite that they believe to be chondrichthyan in origin based on e.g. shape and size and that they will perform EDS and Micro CT scanning to observe it’s contents to try and deduce the producer.
All references mentioned in the reference list are included in the text and vice-versa. There is one small spelling error in Line 100: there should be an accented e on Jimenez i.e. Jiménez.

Professional article structure, figures, tables. Raw data shared.
The article is generally well-structured and well-written, the figures however could do with improvement both in the captions and contents.
Figure 1 -This figure could benefit from additional annotations showing the spiralling shape of the coprolite both externally and internally.
Figure 2 – It is quite confusing – A-D are said to be 3D models, it is only visible that A and B could be 3D models, the figure caption is not clear in describing what C and D are. It would be helpful to the reader if objects of interest where shown in 3D i.e. the brachiopod shell and foram shell (see Qvarnström et al 2019 on filter-feeding pterosaurs for good figure examples)
Figure 3 – This figure would benefit from having images of the coprolite without the crosshair lines present, and also the contrast increased to better visualise the inclusions within the coprolite.
Figure 5 - it would good to also have an image of the teeth from Ptychodus to better support for the author’s claim that the intricate detail of the dentition mean it is was likely a filter-feeder and not durophagous as previously thought.
There are no declarations in the manuscript about where or what raw data is available to the reader i.e. scan data and 3D models. They have included permissions to carry out their field work.

Self-contained with relevant results to hypotheses.
I believe that the authors have done a good job at analysing this coprolite using different techniques like Micro-CT scanning and EDS. However, I believe that the manuscript could benefit from additional analysis and explanation of the dentition of Ptychodus. I feel it would greatly strengthen their conclusions and make the manuscript feel more complete and impactful.

Experimental design

Original primary research within Aims and Scope of the journal.
I am happy that this study lies within the aims and scope of the journal.

Research question well defined, relevant & meaningful. It is stated how research fills an identified knowledge gap.
This has not been addressed well, as I have mentioned previously this could easily be addressed by adding a sentence or two in the introduction.

Rigorous investigation performed to a high technical & ethical standard.
The author's have analysed the coprolite in great detail through EDS and Micro-CT scanning, a couple of points should be clarified further.
Lines 71-72: The author’s describe the coprolite of having a typical size, it would be good to clarify what it is typical of – other coprolites from the deposit or typical in size for a chondrichthyan dropping or other. They also state that the coprolite is incomplete and was likely two times in length, could the author’s perhaps add additional comments to why they believe this and perhaps include an illustration in figure 1 of what they think the complete specimen would have looked like? It would be beneficial if the authors could also include what type of spiral coprolite this is i.e. heteropolar or amphipolar, there may not be enough material to do this so I appreciate it if this is not possible.
It would also be constructive for the authors to mention whether there are other coprolites from their locality, number and approximately the different morphotypes as well as if there are any other coprolites similar in character to the one they have described.

Methods described with sufficient detail & information to replicate.
The authors should include at what resolution the coprolite was scanned at. They have also produced 3D models as seen in Figure 2 and so they should include details of what programme they used and what format the images were imported into the programme (also perhaps how many images were in the image stack and whether they were binned).

Validity of the findings

Impact and novelty not assessed. Meaningful replication encouraged where rationale & benefit to literature is clearly stated.
The author's show the importance of analysing coprolite data for inferring feeding habits that would would miss from studying body fossils alone, however, they should have a little bit more information on typical durophagous dentition vs what is observed in Ptychodus.

All underlying data have been provided; they are robust, statistically sound, & controlled.
The author's should include a statement on whether the scan data and EDS data are available and if so where it could be accessed. If it is not available, due to a desire to work on the scans further, then they should clearly state this.

Conclusions are well stated, linked to original research question & limited to supporting results.
The conclusions of the coprolite and it's contents are well-founded, however in regards to the producer as I have previously mentioned I feel it would be of benefit for the reader (along with strengthening the conclusions) to have a more visual description of the dentition of Ptychodus by including an image of as part of Figure 5.

Additional comments

I commend the author's for their compact study on this spiral coprolite and for using multiple analysis techniques. I feel it aids in our understanding of different feeding strategies and highlighting the importance of studying coprolites. This manuscript will benefit greatly with clearer and informative figures as well as more information on the dentition of Ptychodus.

I would suggest to the author's to alter the title to better reflect their study, something like: Whose poop is this? Late Cretaceous coprolite suggests filter-feeding habit of Ptychodus

---

## Round 0.2 · Minor Revisions

Thank you for addressing our suggestions and previous concerns. The paper has been become easier to follow and of even higher relevance. With the change of title and text, your interpretations are more in line with the provided evidence, but I agree with reviewer 1 and 2 that a clearer designation of speculation and further balancing your interpretations is necessary. I do not consider this a major change in your interpretations and descriptions but rather in the way your discovery is interpreted and its broader implications. I have nothing against speculation, but this should be placed in the discussion and designated as such.

1) Interpretation of diet: I agree with the title change and that your findings expands the diet of Ptychodus with brachiopods and potential other prey items. However, I feel your interpretations of wider change of diet is a possibility but not a certainty (see reviewer 1 and particularly reviewer 2). My recommendation would be to focus on the support for a broader diet for which you provide the first evidence rather than change in feeding strategy to planktotrophy for which you need additional supporting observations (compare reviewer 2).

2) Supporting literature from other shell-crushing sharks: more support is needed for diversity of diet of other shell-crushing sharks which display diverse dental morphologies (Malzahn 1968; Moy-Thomas & Miles 1971; Alexander 1981; Bachmayer & Malzahn 1983; Pollard 1990; Brett & Walker 2002) as well as the typical diet of Ptychodus beyond inoceramids (compare reviewer 2). Previous works has suggested bivalves, ammonites, crustaceans, and small fishes (Amadori et al. 2019, 2020, 2023; Hamm 2020 as listed by reviewer 2): and the fact it has already been considered facultative durophagous by other others (e.g., Shimada et al. 2009 as listed by reviewer 2). Overlap between shelled prey and Ptychodus should likely also be discussed in this context (see reviewer 2). Your results rather seem to support durophagy rather than planktivory which can only be facultative at best. Please revise the text consistently to be in line with the evidence.

3) Formatting and language issues: I feel some additional changes are necessary to make the text clearer to understand (see comments by reviewer 1 and reviewer 3).

4) Figures: some additional visual aids in figures and figure legends are necessary (see reviewer 3).

Please address these as well as all other points raised by the reviewers including those in annotated pdfs.

I look forward to the revised manuscript and seeing this interesting discovery published.

·

Basic reporting

BASIC REPORTING
The English language should be improved to ensure that an international audience can appreciate your paper. I have made many edits. For example, lines 19,21,24,26 and 28 need editing in the abstract.
The text and figures are appropriate. I have made several edits on the pdf.

Experimental design

Adequate.

Validity of the findings

The revised version dials back on Ptychodus being the definite producer (which I think is appropriate) but I have made some edits to emphasize the possibility rather than certainty.
The conclusions are maybe speculative, but speculation is healthy.

Additional comments

I commend the authors on the many revisions they have made to manuscript which have greatly improved it. Overall, I would recommend publishing this manuscript after some revision.

·

Basic reporting

I am glad the authors have improved their study taking into account some of my previous comments, but I am afraid there are still changes to be made. These new changes (see comments in the attached PDF file) include the expanding and rewriting of some sections of the manuscript. I consider these “major revisions” as they concern key interpretations and conclusions provided by the authors. Although the new comments I have included in the reviewed PDF only consist of useful suggestions for the improvement of the submitted study, I would like to emphasize here that these changes are essential in my opinion to make the submitted study ready for publication.

Motivations for such revision are provided below:

First of all, the idea that Ptychodus, a shark displaying very diverse dental morphologies, feeds on a single type of prey (e.g. inoceramids) is obsolete and unrealistic. Numerous works have already discussed this considering Ptychodus as a sort of facultative durophagous (e.g., Shimada et al. 2009). Currently, this group is indeed well known for being able to feed on a wide range of prey, which includes bivalves (e.g., inoceramids, ostreids and rudists), ammonites, crustaceans and even small fishes (e.g., Shimada et al. 2009; Amadori et al. 2019, 2020, 2023; Hamm 2020). The lack of inoceramids in a single coprolite tentatively assigned to Ptychodus is not considered sufficient by this reviewer to support the planktivorous specialization of the entire genus. Direct fossil evidences of predation on inoceramid bivalves are rare, as they are for many other soft / shelled prey items. Evidences of prey-predator dynamics are actually rare and usually indirect in the fossil record. However, the gut contents of some fossil shell-crusher chondrichthyans include fragments of brachiopods (Malzahn 1968; Moy-Thomas & Miles 1971; Alexander 1981; Bachmayer & Malzahn 1983; Pollard 1990; Brett & Walker 2002). Those shell-crusher chondrichthyans also exhibit abraded dentitions as Ptychodus does (see Bachmayer & Malzahn 1983; Shimada 2012; Amadori et al. 2019; Hamm 2020). This predator/prey association would provide good support for the author’s identification of the coprolite producer as Ptychodus (probably the only one able to crush shelled prey within the Cretaceous shark fauna from Opole). However, the coprolite content (e.g., brachiopod remains) presented in the submitted study seems more likely to lean in favor of a confirmation of Ptychodus durophagy, rather than suggesting any different trophic adaptation. Moreover, the European paleogeographical distribution of Ptychodus overlapped with those of its possible shelled prey (including inoceramids) in most Cretaceous epicontinental seas (see Amadori et al. 2023). As admitted by the authors, this also occurred in the area where the examined coprolite came from: “The biota preserved is numerous and consists of ichnofossils, sponges, inoceramids and other bivalves, brachiopods, fish remains, cephalopods, echinoderms, crustaceans, cnidarians, shark coprolites, land flora, and rare marine reptiles” (see “Geology” section of the manuscript submitted; lines 76-78 of the pdf file). With such an availability of food resources and variously modified teeth, I see no plausible reason (plus no evidences) for an adaptation of Ptychodus to planktivory.

For the aforementioned, this reviewer would suggest to the authors that they further amend the submitted study. A possible option would be confirming the mainly durophagous diet for Ptychodus (this reviewer would agree with such identification based on the durophagous diet of this shark) and proposing brachiopods as an additional possible prey. This could be at least for small individuals (juveniles or smaller species). Conversely, larger individuals could have grinded shells of their prey beyond recognition (so they would not be seen in coprolites) or tiny prey were too poor a nourishment for them, as already hypothesized for other fossil durophagous chondrichthyans by Alexander (1981).

Based on the dental morphology of Ptychodus, the availability of prey with and without shells in the Cretaceous environments they inhabited and the abrasions clearly resulting from crushing shelled prey items, the adaptation to planktivory for Ptychodus is rejected by this reviewer as not supported by solid scientific evidence.

Sincerely,

M. Amadori


PS. If the authors wish to refer to some of the studies I have cited in this review (not required from my side, there can be other studies to cite within the relevant literature), I list them below:

Alexander, R. R. (1981). Predation scars preserved in Chesterian brachiopods: probable culprits and evolutionary consequences for the articulates. Journal of Paleontology, 192-203.

Amadori, M., Amalfitano, J., Giusberti, L., Fornaciari, E., Luciani, V., Carnevale, G., & Kriwet, J. (2019). First associated tooth set of a high-cusped Ptychodus (Chondrichthyes, Elasmobranchii) from the Upper Cretaceous of northeastern Italy, and resurrection of Ptychodus altior Agassiz, 1835. Cretaceous Research, 93, 330-345.

Amadori, M., Amalfitano, J., Giusberti, L., Fornaciari, E., Carnevale, G., & Kriwet, J. (2020). The Italian record of the cretaceous shark, Ptychodus latissimus Agassiz, 1835 (Chondrichthyes; Elasmobranchii). PeerJ, 8, e10167.

Amadori, M., Kindlimann, R., Fornaciari, E., Giusberti, L., & Kriwet, J. (2022). A new cuspidate ptychodontid shark (Chondrichthyes; Elasmobranchii), from the Upper Cretaceous of Morocco with comments on tooth functionalities and replacement patterns. Journal of African Earth Sciences, 187, 104440.

Amadori, M., Kovalchuk, O., Barkaszi, Z., Giusberti, L., Kindlimann, R., & Kriwet, J. (2023). A diverse assemblage of Ptychodus species (Elasmobranchii: Ptychodontidae) from the Upper Cretaceous of Ukraine, with comments on possible diversification drivers during the Cenomanian. Cretaceous Research, 151, 105659.

Bachmayer, F., & Malzahn, E. (1983). Der erste Nachweis eines decapoden Krebses im niederrheinischen Kupferschiefer. Annalen des Naturhistorischen Museums in Wien. Serie A für Mineralogie und Petrographie, Geologie und Paläontologie, Anthropologie und Prähistorie, 85/A, 99-106.

Brett, C. E., & Walker, S. E. (2002). Predators and predation in Paleozoic marine environments. The Paleontological Society Papers, 8, 93-118.

Hamm, S. A. (2020). Stratigraphic, geographic and paleoecological distribution of the Late Cretaceous shark genus Ptychodus within the Western Interior Seaway, North America (Vol. 81). New Mexico Museum of Natural History and Science.

Malzahn, E. (1968). Uber neue Funde von Janassa bituminosa (Schloth.) im neiderrheinisschen Zechstein. Geologisch Jarhbuch, 85, 67-96.

Moy-Thomas, J. A., & Miles, R. S. (1971). Palaeozoic Fishes. W. B.Saunders Co., Philadelphia, 259 p.

Pollard, J. E., 1990, Evidence for diet, in: Paleobiology: A Synthesis (D. E. G. Briggs and P. R. Crowther, eds.), Blackwell, Oxford, pp. 362–367.

Shimada, K., Rigsby, C. K., & Kim, S. H. (2009). Partial skull of Late Cretaceous durophagous shark, Ptychodus occidentalis (Elasmobranchii: Ptychodontidae), from Nebraska, USA. Journal of Vertebrate Paleontology, 29(2), 336-349.

Zangerl, R., & Richardson, E. S. (1963). Paleoecological history of two Pennsylvanian black shales. Fieldiana Geological Memoir, 4, 352 pp.

Experimental design

See above.

Validity of the findings

See above.

·

Basic reporting

The English of the manuscript has improved, but there are parts that could be improved further (I have identified these in an annotated pdf). The authors have amended the references that I had mentioned in my review previously and I am happy with their work on this. The Figures have been greatly improved from before, there are still a couple of small edits to be made which I have highlighted in the pdf.
The authors have added additional information on the teeth of Ptychodus which strengthens their conclusions.

Experimental design

The authors have greatly improved on the points I raised earlier concerning the expansion of the introduction and the description of the coprolite as well as other coprolites from the assemblage, the picture is much clearer. The authors have also included more details on the methods and programmes used and have included a repository to access the data.

Validity of the findings

As aforementioned, access to data has now been provided and they have expanded their description of the teeth of Ptchodus and considered alternative considerations for the coprolite producer. Some small edits are still needed (see annotated pdf).

Additional comments

The authors have done a great job addressing the issues raised by the reviewers, some minor edits are still needed (see annotated pdf) but I am happy to recommend that the manuscript be accepted pending minor edits, I do not think the manuscript should be sent to review again.

---

## Round 0.3 · Minor Revisions

Thank you for making the revisions which have made the manuscript even easier to follow and consistent. I feel the manuscript can be as good as accepted. However, there were some minor inconsistencies remaining which might change the main message of the paper and need to be resolved before publication. It mainly concerns a sentence in the abstract which could be misunderstood that brachiopods are mollusks and your definitions of durophagy and filter feeders (see annotated pdf).

I feel these inconsistencies highligted in the annotated pdf are easy to resolve and i look forward to seeing this published.

---

## Round 0.4 · accepted · Accept

Thank you for addressing these additional suggestions which make the manuscript more consistent and easier to follow. I am happy to be able to accept the manuscript and look forward to seeing it published. I found an additional typographical issue and have one additional suggestion for the abstract for you to consider for the sake of clarity during the proofing phase. Please see the attached file.